# Unveiling endothelial cell border heterogeneity: VE-cadherin adherens junction stratification by deep convolutional neural networks

**Rudmer J. Postma**[1]*, **Susan E. Fischer**[2], **Roel Bijkerk**[1], **Anton Jan van Zonneveld**[1]

**1** Department of Internal Medicine, Leiden University Medical Center, Leiden, The Netherlands,
**2** Department of Gastroenterology and Hepatology, Leiden University Medical Center, Leiden, The Netherlands

* r.j.postma@lumc.nl

## Abstract

### Background

Systemic diseases are often associated with endothelial cell (EC) dysfunction. A key function of ECs is to maintain the barrier between the blood and the interstitial space. The integrity of the endothelial cell barrier is maintained by VE-Cadherin homophilic interactions between adjacent cells. The morphology of these borders is highly dynamic and can be actively remodeled by numerous drivers in a (patho)physiologic context specific fashion.

### Objectives

High-content screening of the impact of circulatory factors on the morphology of VE-Cadherin borders in endothelial monolayers *in vitro* will enable the assessment of the progression of systemic vascular disease. We therefore aimed to create an image analysis pipeline, capable of automatically analyzing images from large scale screenings, both capturing all VE-cadherin phenotypes present in a sample while preserving the higher-level 2D structure. Our pipeline is aimed at creating 1D tensor representations of the VE-cadherin adherence junction structure and negate the need for normalization.

### Method

An image analysis pipeline, with at the center a convolution neural network was developed. The deep neural network was trained using examples of distinct VE-Cadherin morphologies from many experiments. The generalizability of the model was extensively tested in independent experiments, before further validation using ECs exposed *ex vivo* to plasma from patients with liver cirrhosis and proven vascular complications.

### Results

Our workflow was able to detect and stratify many of the different VE-Cadherin morphologies present within the datasets and produced similar results within independent

**Data Availability Statement:** Data, models, and scripts are available on the zenodo repository: https://doi.org/10.5281/zenodo.13936923.

**Funding:** The author(s) received no specific funding for this work.

**Competing interests:** The authors have declared that no competing interests exist.

experiments, proving the generality of the model. Finally, by EC-cell border morphology profiling, our pipeline enabled the stratification of liver cirrhosis patients and associated patient-specific morphological cell border changes to responses elicited by known inflammatory factors.

## Conclusion

We developed an image analysis pipeline, capable of intuitively and robustly stratifying all VE-Cadherin morphologies within a sample. Subsequent VE-Cadherin morphological profiles can be used to compare between stimuli, small molecule screenings, or assess disease progression.

## Introduction

Endothelial cells (ECs) line the inner surface of the blood vessels. Here, they function as a barrier between the plasma and the interstitial space, control vascular permeability and facilitate the innate immune response [1,2]. Circulating plasma factors such as pro-inflammatory cytokines or histamine can activate EC receptors and cause reversible alteration of the EC barrier permeability [3]. EC dysfunction and impaired EC barrier permeability play a key role in the pathophysiology of many disease, including sepsis, cardiovascular disease, diabetes, chronic kidney disease, and SARS-CoV2 related complications [4–9].

The integrity of the EC barrier is maintained by various types of cell-cell junctions; adherens junctions (AJs), gap junctions, and tight junctions, which are remodeled upon EC activation [10]. Tight junctions and adherens junctions are closely associated in ECs, where tight junctions regulate monolayer permeability, adherens junctions maintain cell–cell adhesion, regulate monolayer permeability, and play a role in actin cytoskeleton remodeling [11]. The homophilic interactions of VE-cadherin(CD144) in the adherens juctions are the major determinant of endothelial barrier integrity and can display distinct morphologies [12]. In non-activated ECs, VE-cadherin is linearly distributed along cell borders, but also forms reticular structures that cover significant areas at the cell borders [13]. Following activation of ECs by TNF-α, histamine, or by plasma derived from COVID-19 patients VE-cadherin AJs display less reticular structures, but do display disrupted linear areas, and many areas of reduced border thickness [13,14].

Computational tools for analyzing microscopy images have proven to be highly useful for the assessment of VE-cadherin morphological heterogeneity. Several methods exist for analyzing VE-cadherin AJs. These methods can be divided into two groups; colocalization of VE-cadherin signal with F-actin to determine focal adhesion points and distinguish phenotypes [15,16], and the segmentation based tools, with CellBorderTracker by Seebach et al [17] being the most notable example, allowing to quantify border continuity, gaps, and cross-sectioning.

These methods and tools are focused on analyzing just a few borders and cells, and are therefore not suited for analyzing images obtained from large scale screenings. Moreover, translating the output of these methods into 1D datapoints capturing the often subtle phenotypes of VE-cadherin AJ morphology is cumbersome at best, and higher-level 2D structure is lost by these methods.

Here, we present a different methodology to analyze VE-cadherin AJs, with the goal to develop a tool suitable for analyzing images from large scale screenings, both capturing all VE-cadherin AJs phenotypes present in a sample while preserving the higher-level 2D structure.

This method is aimed at creating 1D tensor representations of the VE-cadherin AJ structure by using deep convolutional neural networks, that can be used for further data analysis.

## Methods

### Cell culture

Human umbilical vein endothelial cells (HUVECs) were isolated from umbilical cords and cryogenically frozen until use, as previously described [18]. HUVECs were cultured in a 1% gelatin (Merck; 104078) coated T75 flask in endothelial cell growth medium 2 (EGM-2) (PromoCell C-22111) supplemented with 1% Penicillin/Streptomycin (Gibco 15070–063). Cells were transferred to a 1% gelatin coated, thin bottom 96-wells (PerkinElmer Phenoplate, 6055302) 4 days prior to stimulus exposure to allow a cobblestone-like confluent monolayer to form.

### TNF-α, IL1-β, VEGF, Thrombin, and Histamine stimulation

Confluent EC monolayers were exposed to either 0.1 or 10 ng/ml tumor necrosis factor alpha (TNF-α) (Sigma-Aldrich H8916) overnight, 100 ng/ml $VEGF_{165}$ (Peprotech, 100–20) either overnight or for two hours, 1 ng/ml or 10 ng/ml IL1β (Peprotech, 200-01B) overnight, 50 μM Histamine (Sigma Aldrich H7125) for 2 hours, 10 ng/ml TNFα + 10 ng/ml TGFβ1 (Peprotech 100–21) overnight, or Thrombin for 2 hours. All treatments were performed in endothelial cell basal medium, supplemented with 1% Penicillin/Streptomycin (Gibco 15070–063) and 2% FCS.

### Patient-derived plasma and ethics statement

For this study, high-content image datasets obtained in the study of Postma et al, [19] were reanalysed. This pilot study into endothelial cell dysfunction and endothelial cell phenotyping was a retrospective study using stored EDTA plasma samples in a local biobank. The use of these samples for endothelial cell phenotyping was approved by the Liver Diseases Biobank and by the Medical Ethics Committee MDL; "Medisch-ethische toetsingscommissie Leiden Den Haag Delft" (064/SH/sh;3.4120/09/FB/jr; METC LDD-number: B21.014). All patients gave written informed consent for inclusion in the biobank and for their medical records to be used in research, and no minors were included. Authors of the current manuscript had no access to information that could identify individual participants, and original biological samples were accessed on 12-10-2021. EDTA samples from two different patient groups were selected: (i) patients with compensated cirrhosis and (ii) patients with decompensated cirrhosis. Diagnosis of cirrhosis was based on histology, imaging and/or laboratory results.

### Immunofluorescent staining and imaging

Cells were fixed and nuclei and VE-Cadherin were stained as previously described [19]. In short, cells were fixed with 4% paraformaldehyde (Alfa Aesar J61899) in PBS for 10 minutes, washed with blocking buffer (2% (w/v) BSA (Sigma A7030), 0,5% (w/v) Glycine (Merck 1042011000), 1% (v/v) Triton X-100 in PBS (Gibco 10010023), followed by a 20 minute incubation with blocking buffer. Primary antibodies to VE-Cadherin (BD 555661, 2 μg/ml in PBS + 2% (w/v) BSA) were incubated for one hour. Cells were washed three times with PBS containing 2% (w/v) BSA and subsequently incubated with 488 Alexa-fluorophore labeled Donkey anti-mouse antibodies (Invitrogen A-11001, 2 μg/ml) and 1:1000 HOECHST 33258 (Molecular Probes) in PBS + 2% BSA for one hour. Cells were washed with PBS and stored under Borate Buffer, 50mM, pH = 8.3.

Max-projections of 9 z-steps with 0.5 μm step size were acquired using a high content confocal microscope (Molecular Devices, ImageXpress™ Micro Confocal) at 20x magnification (Nikon Plan Apo Lambda; NA = 0.75), using a 60μm pinhole. Dapi and FITC channels were used to acquire nuclei and VE-Cadherin, respectively. Nine sites without overlap were imaged per well.

Images were checked for focus and the presence of outliers according to [19–21]. In short, visual inspection of outliers (images that contained artifacts; out-of-focus images, debris, clipping/saturation artifacts) was performed before being discarded. Sites that contained failed images in a single channel were discarded altogether.

Following image quality control, images were corrected for uneven illumination and vignetting.

In short, a focus-score metric, total sum, median, standard deviation, and the 0.01, 0.25, 0.75, 0.99 quantiles of the intensity values were obtained for each image [20,21]. Results were plotted in histograms comprised of all images for a given channel identify outliers [21] followed by inspection of images to assess image quality before discarding. Images that contained artifacts (e.g. out-of-focus images, debris, clipping/saturation artifacts), were discarded. Sites that contained failed images in a single channel were discarded altogether.

Images were corrected for uneven illumination and vignetting by using an approach, based on Draper et al. [22]. Bright areas or debris were iteratively filtered out by assigning areas with intensity values 3*SD above the local median pixel value the intensity value of the direct neighborhood, after which a median weighted average of the images was computed, followed by smoothing by a Gaussian filter. The image used for correcting illumination was constructed by dividing the mean background value by the smoothed artificial background image. All images in the corresponding channel and plate were multiplied by the correction image to obtain the illumination corrected images.

## Image analysis pipeline, convolutional neural network development, and statistical analysis

Python 3.10.9, running the packages cv2 (version 4.7.0), skimage (version 0.19.3), numpy (version 1.23.5), umap-learn (version 0.5.3), seaborn (version 0.11.2), tensorflow (version 2.10.1), scikit-learn (version 1.5), hdbscan, pickle, and keras (version 2.9.0) was used to develop and run the image analysis pipeline and subsequent analysis.

In essence, the pipeline consists of first identifying cell borders via seeded segmentation, using the nuclei as seeds. Afterwards, regions of interest -a square of 64x64 pixels with the center at the cell border- were sampled and stored for analysis or for the creation of a training/validation dataset.

Nuclei were identified from HOECHST images by first normalizing the image to the 0.01 and 0.99 quantile, followed by applying a gaussian filter with a disc shaped kernel with 5 pixel diameter. Subsequently, images were binarized using the OTSU threshold, followed by opening and closing morphological operations. Individual nuclei were separated using watershed.

Cells were identified from VE-cadherin images by first normalizing the image to the 0.01 and 0.99 quantile, followed by applying a median filter with a disc shaped kernel with 5 pixel diameter to smooth the image. A gradient filter was applied to find VE-cadherin edges, afterwards a seeded watershed was applied to segment and identify cells and cell borders.

A Convolutional Neural Network (CNN) architecture was built, optimized, and executed using the Keras and TensorFlow python packages, with CUDA 11.8. The model was trained with batch-size 64, for 100 epochs on lr = $1*10^{-4}$, and the model best performing on the validation set at epoch 40 was chosen. Subsequently, a new model was created for extracting

image embeddings. Images were fed into this model, and embeddings of the images were extracted for further analysis.

The differences between population of the clusters for different conditions were compared using Hotelling's T-squared test on a LDA reduced dataset, using the first two components, explaining 88% of all variance, this in order to be able to include the spread in distributions from technical replicates. To determine if the distribution between biological replicates do not differ statistically, the Hotelling's T-squared test was used, using a LDA model calculated over only the first biological replicate, and fitting both biological replicates into this pre-fitted model, observing non-rejection as a means for that the distributions do not differ significantly. To compare differences between the profiles from healthy controls, compensated liver cirrhosis patients, and decompensated liver cirrhosis patients, first an LDA model was constructed to reduce the dataset to a singular dimension. Afterwards, groups were compared by ANOVA and t-test. In addition, cosine similarity was computed between the datapoints for visualizing the similarity by heatmap. Cosine similarity as a metric was chosen because of its well described use in comparing morphological data in the field of high-content screenings [23], and because the cluster-profiles represent a frequency vector, and therefore the approach for comparing frequency vectors by cosine similarity was adopted from the field of text analysis [24].

## Results

### VE-cadherin adherens junction morphology changes depending on the stimulus

To obtain multiple datasets containing large amounts of examples of different VE-cadherin AJ morphologies, ECs were stimulated with varying concentrations and exposure times of either TNFα, IL1β, VEGF$_{165}$, TNFα + TGFβ1, histamine, or thrombin. TNFα and IL1β are pro-inflammatory cytokines involved in several disease and conditions including sepsis and tumor invasion, and both cytokines induce degradation of the EC barrier and expression of adhesion molecules [25,26]. The growth factor VEGF is crucial for angiogenesis both physiological in wound healing as well as pathophysiological in many conditions including cancer and diabetic complications [27]. EC stimulation by histamine or thrombin causes acute degradation of the EC barrier and is involved in allergic reactions and coagulation events, respectively. Stimulation by TNFα + TGFβ1 is often used as a model for Endothelial-Mesenchymal Transition (EndoMT) in the pathogenesis of fibrotic diseases [28].

Visual examination of the VE-cadherin AJ morphology clearly showed that different stimuli elicit different morphological responses (Fig 1). While unstimulated controls show mature borders and reticular areas (Fig 1A), stimulation by either VEGF$_{165}$ or Histamine showed formation of focal adhesions, resulting in zig-zag like border morphology (Fig 1B and 1C). TNFα and IL1β induced similar disturbed morphologies in a dose dependent manner (Fig 1D and 1F). Interestingly, VE-cadherin AJ morphology induced by stimulation by both TNFα and TGFβ1 differed from the morphology of stimulation by TNFα only, as more deterioration of the borders was observed together with more preservation of the reticular areas (Fig 1H).

### Pipeline and deep convolutional neural network development

We constructed a computational pipeline aimed at providing an intuitive and robust overview of all observed VE-cadherin morphologies present in the microscopy images within a dataset (Fig 2A). Different VE-Cadherin morphologies were visualized as datapoints in a UMAP allowing for easy interpretation of the results. Combined with clustering algorithms this

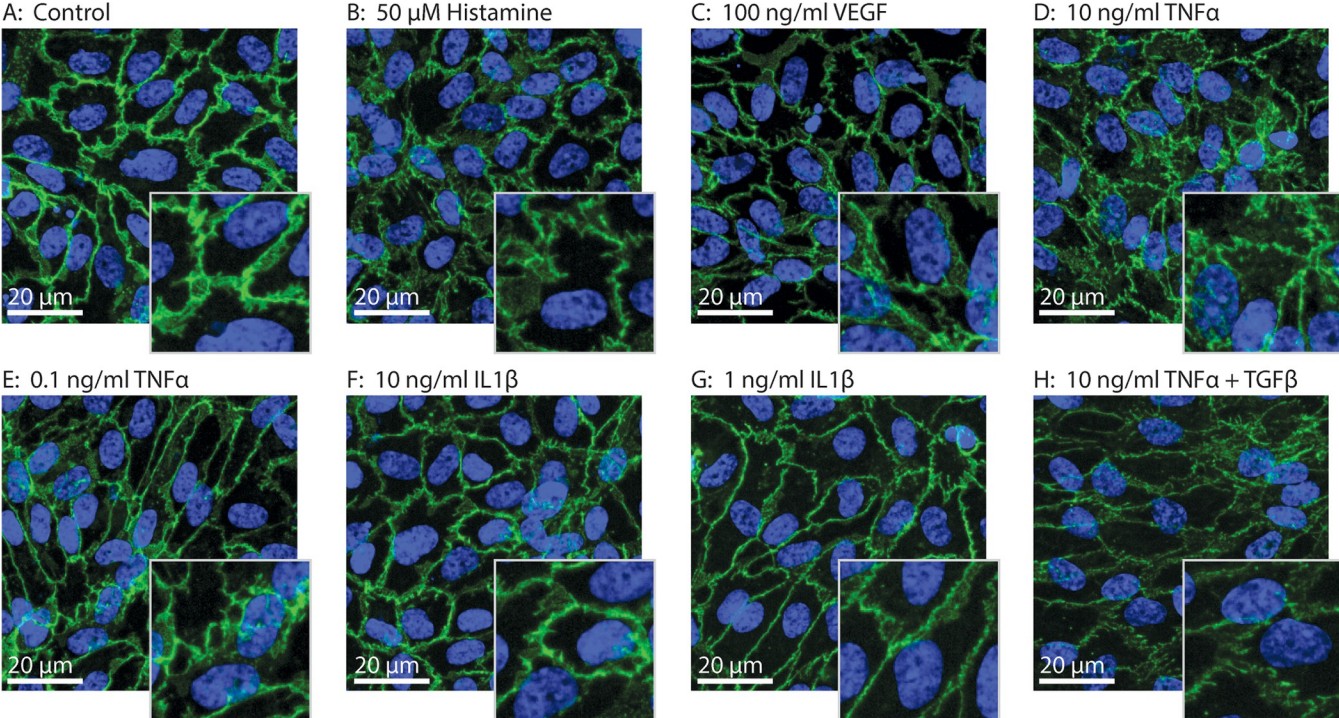

**Fig 1. VE-Cadherin adherens junction morphology.** VE-Cadherin AJ morphology changes upon EC stimulation depending on the stimulus. Stimulation by histamine induces a zig-zag appearance of the border morphology, while stimulation with TNFα induces a narrow border phenotype. Images were acquired using a high quality 20x magnification lens.

approach allows for easy comparison and visualization between different experimental conditions. Border morphologies were encoded as datapoints by taking regions of interest (ROIs); 64x64 pixel images of the border, and using a trained deep convolutional neural network (CNN) to return an embedding of the ROI.

The main objective of the model architecture was to yield a low dimensional embedding of input image, capturing enough information about the VE-cadherin border morphology to allow for stratification of morphology. A modification of the Xception [29] deep convolutional neural network architecture was used, simplified and reduced in dimensions to accommodate the image size, improve generalization, and to yield an 32-dimensional vector embedding of the original input image (Fig 2B). This embedding is the activation of a Global Pooling layer, thereby destroying information about the localization or rotation of the borders within the ROI, while retaining information about the morphology and extent thereof present in the ROI. The network was designed to solve a 5-class classification tasks; to classify images into the five main VE-cadherin morphologies: highly disturbed areas, disturbed borders, linear borders, foci, and reticular areas (Fig 2C). As the filters/weights of the model are trained to distinguish between different VE-cadherin morphologies, this classification task serves as a proxy for the creation of image embeddings. A training/validation dataset was composed by hand to contain numerous examples of each morphology, obtained from 6 independent experiments conducted using different HUVEC donors (Fig 2C). Augmentation of training dataset, by random vertical and horizontal mirroring, random brightness adjustment, and random contrast adjustments of the image was used to increase the size of training dataset and improve generalization. The model was trained for 100 epochs, and the model with best validation loss and accuracy was chosen (epoch 40).

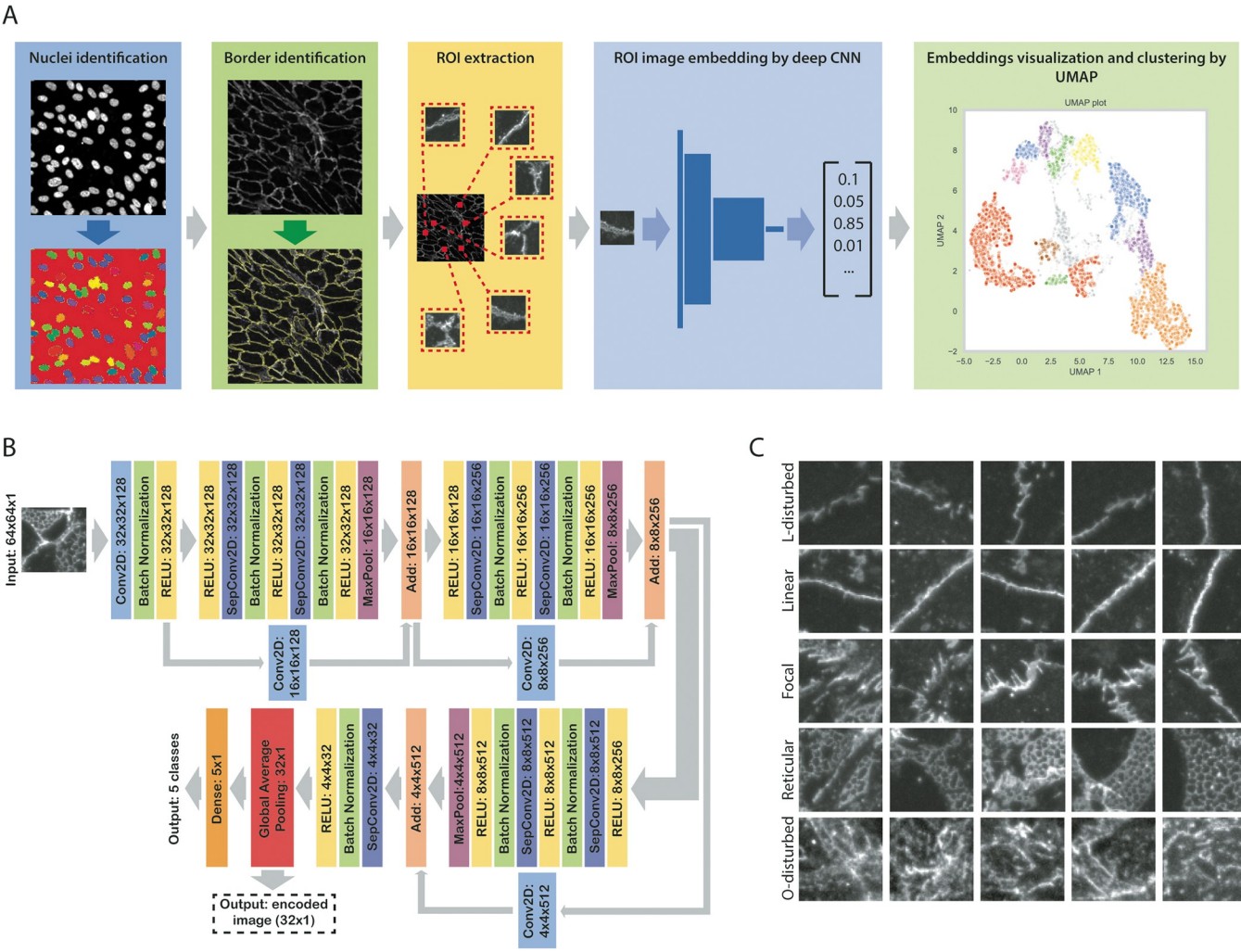

**Fig 2. Analysis pipeline overview. A:** The VE-cadherin AJ border morphological quantification pipeline consist of first identifying the nuclei in normalized and illumination corrected DAPI images, followed by identification of VE-cadherin borders, using the nuclei as seeds. 10,000 64x64 pixel images (regions of interest (ROIs)) were then extracted from normalized and illumination corrected VE-Cadherin images, with the cell borders as the center of these ROI images. These images were fed into the deep CNN to create embeddings of the images. These embeddings were then used as datapoints to construct a UMAP for dimension reduction, visualization, and clustering. **B:** Model architecture of the deep neural network, resulting in both classification and embedding of the input image. **C:** examples of the training categories used for the proxy classification task.

## VE-cadherin AJ morphology is resembled in the embeddings

After training of the network, a new set of VE-cadherin ROI images was extracted from a microscopy image set containing examples of ECs exposed to known stimuli; TNFα, IL1β, Thrombin, Histamine, and VEGF. These images were fed into the network, and embeddings of these images were extracted from the activation of the "Global Average Pool" layer within the model (Fig 2B), giving a 32-dimensional vector for each image that can be used as a single datapoint for further analysis. Embeddings were visualized and reduced in dimension by UMAP, fitted over 10,000 VE-cadherin ROI image examples obtained from multiple examples of all stimuli.

Multiple clusters could be identified by the HDBSCAN algorithm in the 2-dimensional UMAP. In order to link VE-cadherin morphology to the clusters, and to validate that the clusters indeed correspond to different morphologies, five images were extracted for at random

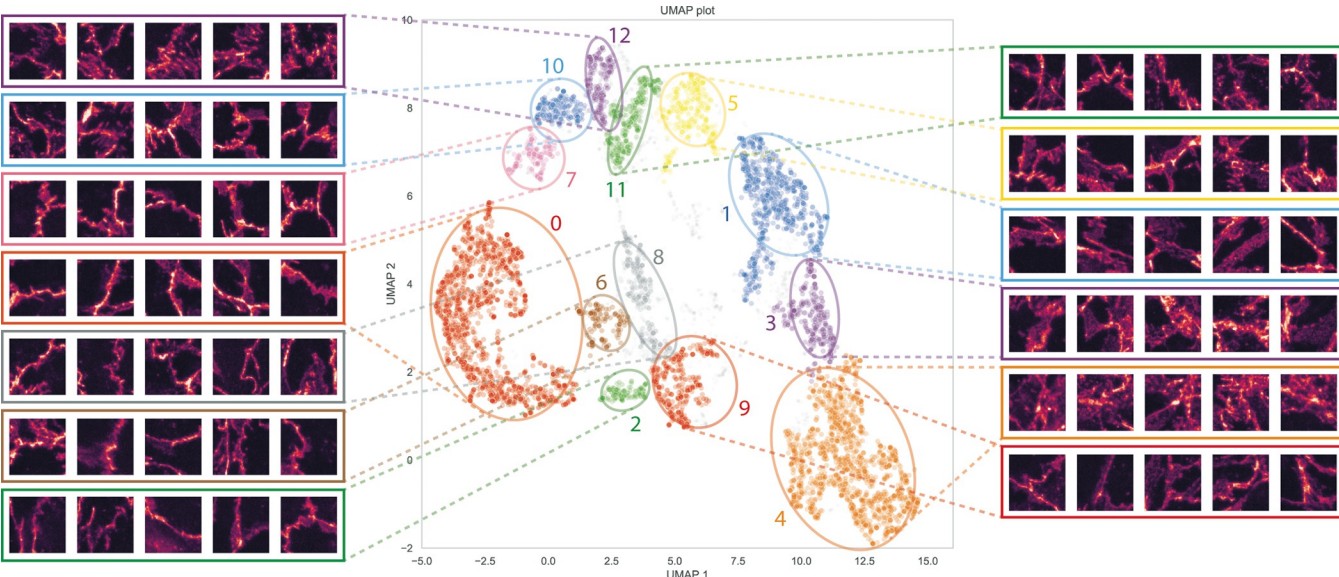

**Fig 3. Overview of VE-Cadherens morphology per cluster.** Embedding reduced in dimension and visualized by UMAP. Clusters correspond to different VE-Cadherin AJ border morphology. For clarity's sake the cluster enumeration corresponds to the results from the HDBSCAN algorithm.

for the bigger clusters (Fig 3). Visual comparison between the images showed that different clusters indeed correspond to different morphologies and different morphologies cluster together.

Cluster enumeration was kept from the HDBSCAN results. Visual investigation of the borders shows that cluster 0 mostly contains quiescent mature borders. Clusters 6, 2 and 8 contain mostly disrupted thinning borders. Cluster 9 mostly contains thinning borders with additional zig-zag morphology. Clusters 7, 10, and 12 contain increasingly disturbed borders with increasing amounts zig-zag/focal adhesion point morphology. Clusters 1, 5, and 11 contain reticular junctions, increasingly disturbed in cluster 5 and 11 respectively, eventually resulting in a more focal adhesion point-like morphology. Cluster 3 and 4 show increasingly disturbed borders without the presence of focal adhesion points, but with a more diffuse, spread out, highly disturbed border, corresponding to a morphology observed after high-dose TNFα stimulation.

## Stratification between different stimuli

To compare VE-cadherin morphology between different conditions, a larger set of 6000 VE-cadherin ROI images was extracted for each stimulus from multiple wells (technical replicates), fed into neural network, and the resulting embeddings were transformed by the previously fitted UMAP. This allowed the comparison between the different conditions by comparing UMAPs and cluster occupancy (Fig 4) and test if the cluster occupancies are indeed statistically different. LDA analysis showed major shifts in distribution of morphologies for some stimuli (Fig 4), while showing adequate grouping of the technical replicates. Major differences in morphology distribution between the unstimulated control and 10 ng/ml TNFα could be observed, where stimulation by 10 ng/ml TNFα resulted in almost complete loss of cluster 0, 1, 5, 7, 10, and 11, and a shift towards cluster 9 and 4, representing highly disturbed morphology ($P < 0.005$). Contrary, when stimulating cells by a combination of TNFα + TGFβ there was an enrichment observed for clusters 11 and 12 compared to the TNFα only condition ($P < 0.05$). Exposure to 0.1 ng/ml TNFα resulted in a slight enrichment of cluster 3 and 4

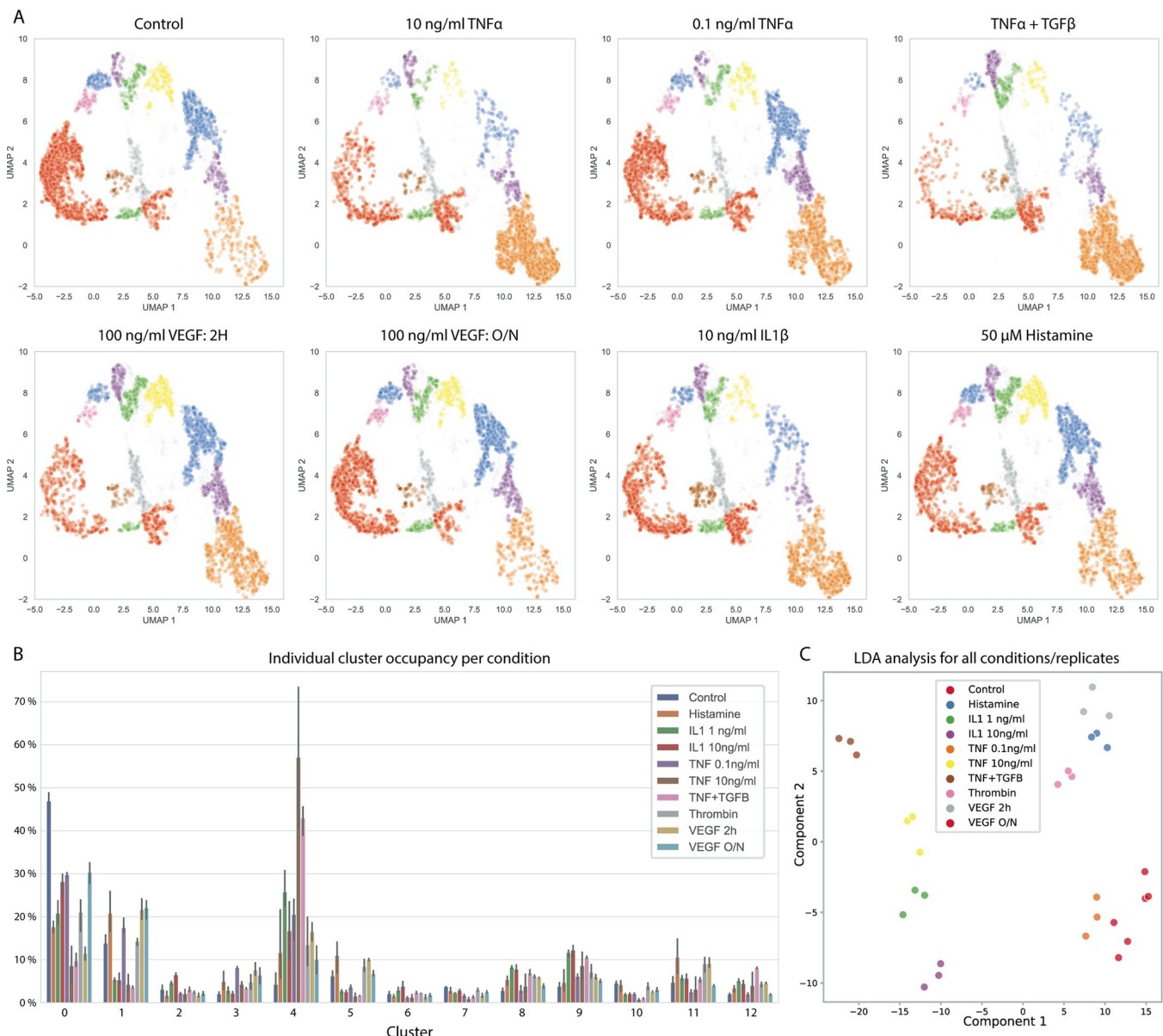

**Fig 4. Cluster occupancy per stimulus. A:** Example UMAP visualizations of VE-Cadherin border morphologies for different stimuli for all technical replicates. **B:** Each different stimulus results in shifts in cluster occupancy, as is displayed by the cluster occupancy per condition (bars indicated standard deviation between three replicate conditions). **C:** LDA analysis explaining 70% and 18% of variance for Component 1 and 2, respectively, shows the distribution between different stimuli, and the clustering of replicates conditions in the dimension reduced space.

from cluster 0 compared to control (P < 0.05). Stimulation by high-dose IL1β resulted in a similar shift compared to the unstimulated condition, though less severe than stimulation by TNFα (P < 0.05). Stimulation by 50 μM Histamine resulted in zig-zag focal borders to become a dominant phenotype marked by a shift towards cluster 2, 5, 6–12, and a loss in cluster 0 (P < 0.05). Interestingly, stimulation of ECs by 100 ng/ml VEGF for two hours, resulted in different distribution of morphologies when compared to overnight exposure, where two hour exposure resulted in gain in 2, 5, 6–12, and a loss in cluster 0 compared to no exposure (P < 0.05), overnight exposure resulted in a phenotype close to the control condition

(P < 0.05), and showed loss 2,6,8,9 when compared to the two hour stimulation (P < 0.05). Stimulation by 50 μM Histamine resulted in similar phenotype compared to stimulation by 100 ng/ml VEGF for two hours (P = 0.12).

## Generalizability validation of the deep CNN

To investigate the generalizability of the network and if similar cluster occupancies are obtained between different experiments, an independent experiment was performed using HUVECs isolated from a different donor not included in the training dataset, but activated by a selection of the same stimuli. Again, VE-cadherin ROI images were extracted for each stimulus with multiple technical replicates combined and fed into network. The resulting embeddings were transformed by the pre-fitted UMAP and cluster occupancy was determined by predicting cluster belonging using the previously fitted HDBSCAN model. Comparing mean UMAPs between the same stimulus and condition showed similar distribution of the VE-Cadherin morphologies (Fig 5A and 5B). Using an LDA model computed using only the dataset described above on the selected conditions, cluster belongings of this new dataset were transformed and compared by Hotelling's t-squared test between the same stimuli for this

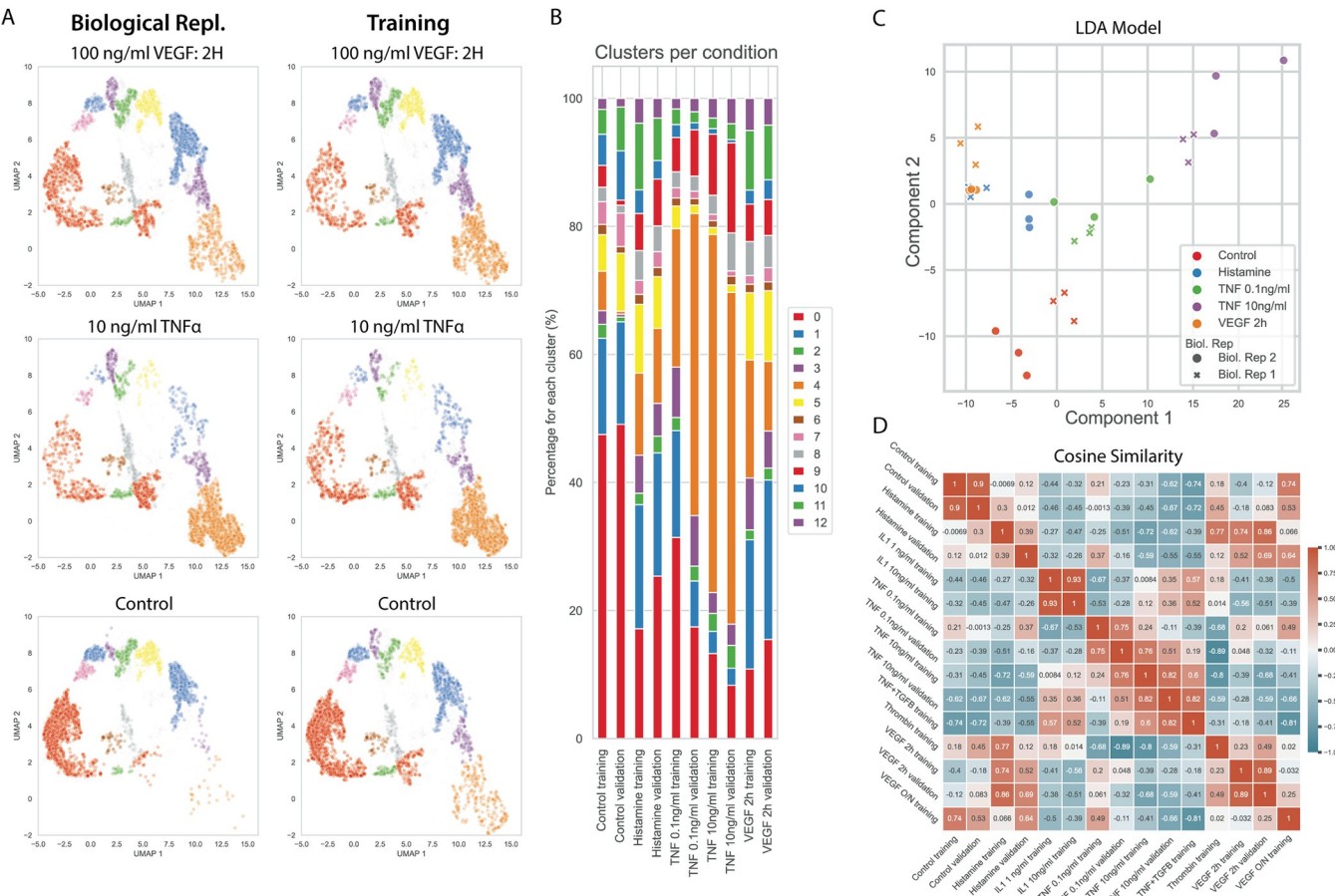

**Fig 5. Similarity between biological replicates. A:** morphology embeddings visualized by UMAPs showed that clusters occupancy is comparable between one of the experiments used in training of the deep CNN, and a completely independently performed experiment. **B:** Mean occupancy of each cluster per condition. **C:** LDA analysis of the replicates of both experiments transformed using a model calculated from the data of Fig 4 for these stimuli. **D:** Heatmap showing the cosine similarity between the one of the training experiments and the independently performed validation experiment (mean profile used per condition).

independent experiment, and experiment described above, Fig 5C. In addition, the similarity between the mean profiles were visualized as a cosine similarity matrix, Fig 5D.

Stimuli showed comparable cluster distributions between the training experiment and the validation experiment, with cosine similarity scores ranging from 0.4 to 0.9 depending on the stimulus and concentration, and close proximity within the LDA plot between different conditions. Unstimulated control conditions from the trainings set vs unstimulated control conditions in the independent biological replicate showed a cosine similarity of 0.9, with close proximity of the both biological replicates in the LDA plot, however, cluster occupancy seemed to be slightly different between both replicates P = 0.02, Fig 5C. Histamine showed a cosine similarity of 0.39 between the two datasets, and both replicates clustered near each other in the LDA plot, however with a distinct seperation between the two P = 0.008. TNFα stimulation showed similar cluster distributions between both experiments with a cosine similarity of 0.75 and 0.82 with P = 0.12 and P = 0.25 for 0.1 ng/ml and 10 ng/ml TNFα respectively. VEGF exposure for two hours showed similar cluster distributions between both experiments, with a cosine similarity of 0.89 (P = 0.08).

## VE-cadherin stratification of EC morphological responses to patient-derived plasma

Recently, we demonstrated that for liver cirrhosis disease-specific endothelial cell phenotypes can be induced *ex vivo* by exposing ECs to patient-derived EDTA plasma, and that high-content screening is a valuable tool to obtain this phenotypical information [19]. To investigate whether VE-Cadherin patient specific morphological profiles can be extracted using the deep-learning methodology presented here, a previously acquired high-content dataset [19] was reanalyzed for VE-Cadherin morphology.

Cluster belongings, again computed using the pre-fitted UMAP and DBSCAN model, were computed for EC exposed to plasma from an individual patient or healthy control and subsequently used as that patient or healthy control its specific profile. PCA analysis of the resulting profiles showed that VE-Cadherin morphology distribution changes with the disease status of liver cirrhosis; which is divided into a compensated, asymptomatic phase and a decompensated, symptomatic phase [30] (Fig 6A). VE-Cadherin morphological profiles obtained from exposing healthy control plasma to ECs clustered together and showed some overlap with the profiles obtained from the least severely affected liver cirrhosis patients (P < 0.05). Decompensated patients, the most severely affected patients, showed a large divergence from healthy control (P < 0.0001), and showing again some overlap with the compensated group (P < 0.0001). Visual inspection of the original images indeed confirmed that VE-Cadherin morphology is altered depending on the disease severity, with decompensated liver cirrhosis having the most severe phenotype (Fig 6C).

## Discussion

In this study, we presented a fully automated pipeline for the analysis of high-throughput screening of endothelial cell VE-Cadherin morphology. As a proof-of-concept we showed that our pipeline is capable of identifying subtle changes in VE-Cadherin morphologies given known EC stimuli. In addition, we showed our pipeline is able to stratify VE-Cadherin morphologies induced by exposing ECs to patient-derived plasma, thereby generating patient-specific VE-Cadherin morphological profiles.

Many systemic diseases are connected to endothelial cell dysfunction, including sepsis, cardiovascular disease, diabetes, chronic kidney disease, and SARS-CoV2 related complications [4–9]. VE-Cadherin morphology is deeply connected to endothelial cell function, and its

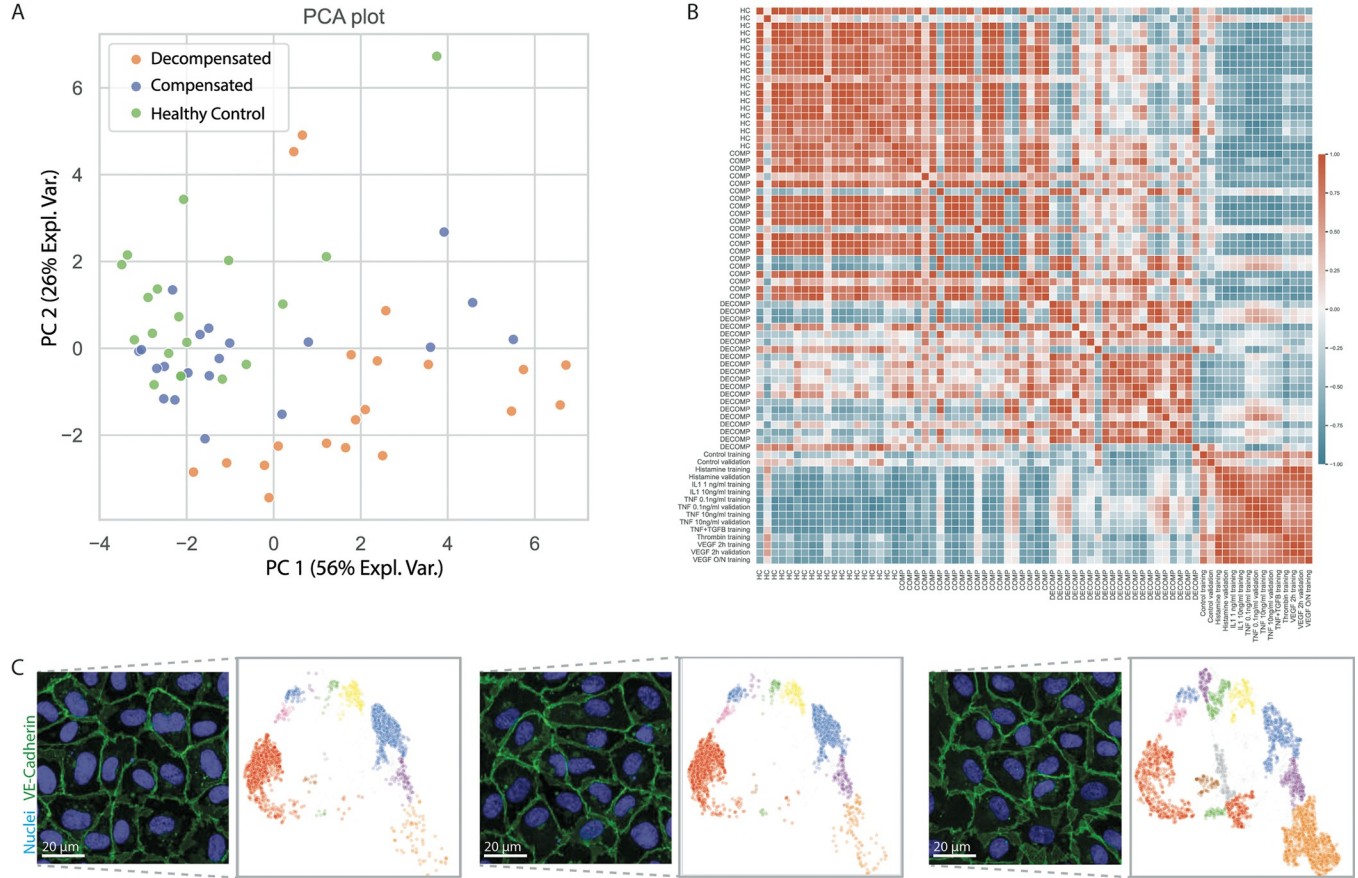

**Fig 6. Analysis of VE-Cadherin borders of EC exposed to liver cirrhosis patients. A:** PCA plot of the VE-cadherin cluster profiles of ECs exposed to patient derived plasmas or healthy control plasmas. Plotted are the first and third PC, both containing information about the disease severity, explaining 79% of variance combined. **B:** Heatmap showing the cosine similarity between patient profiles and the VE-Cadherin morphologies induced by EC stimulants. Color corresponds to cosine similarity score with the red color indicating high, and the blue color indicating low similarity. **C:** Examples of cell morphology for ECs exposed to plasma from either healthy controls, compensated liver cirrhosis patients, or decompensated liver cirrhosis patients. ECs of decompensated group are more highly activated compared to healthy controls.

morphology determines the cell barrier function [31,32]. Here, we showed that VE-cadherin morphology is highly heterogeneous, and that the distribution of different morphologies is dependent upon the stimulus and its concentration, or upon the circulating factors present in patient-derived plasma. Our pipeline provides a valuable tool to gain better insight into the different EC responses to different classes of stimuli.

Our pipeline was able to discern between the different VE-cadherin morphologies, and capture this information in clusters of VE-Cadherin morphology that can be used in further data analysis as VE-Cadherin "morphological profiles"; a 1D vector containing the heterogeneity of all VE-Cadherin morphologies present in a sample. Using these profiles we were able to stratify the different stimuli. As expected, the unstimulated control condition showed higher population of clusters 0 and 1 compared to the highly stimulated conditions, representing mature borders and mature reticular areas respectively, where the 10 ng/ml TNFα condition showed highly disturbed areas, represented by cluster 3 and 4. As stimulation by high dose TNFα is associated with decreased endothelial cell border integrity [14], this specific morphology potentially corresponds to leaky borders. In addition, enrichment of clusters 5, 8,9, 11, and 12 as observed for stimulation by histamine likely is also associated with increased barrier

permeability, as stimulation by histamine is known to induce a leaky EC phenotype [14]. Interestingly, we also observed a time dependent effect of VEGF. As previously reported [33,34], we observed more zig-zag like remodeling of the VE-Cadherin borders after two hours of VEGF exposure, compared to more slightly disturbed morphology after overnight exposure to VEGF. This information was also well represented in the change in clustering, and thus in the VE-Cadherin morphological profiles.

We have shown our pipeline is highly generalizable, capable of generating similar profiles between two independent experiments performed with different EC donors and cell seeding density. Classical image analysis often suffers from plate, experimental batch, and EC donor effects, and common strategies involve Z-score normalization of the data to a common internal standard [21,35,36]. Using a convolutional neural network instead of using handcrafted features allows for discovery of relevant patterns without the measurements being predefined, and potentially negates the need for normalization. By learning relevant features over multiple datasets, with different EC donors, the network potentially learns generalizable patterns, and thus might prevent the plate, batch, and donor effects commonly observed when working with high-content data. This allows easy comparison between completely independent experiments without the need of normalization, and thus enables comparison with stored VE-Cadherin morphological fingerprints from known stimuli, acquired in independent screenings.

In a recent publication, we showed that high-content screening of ECs stimulated by EDTA plasma derived from liver cirrhosis patients allows for patient stratification based on EC morphological responses. Systemic inflammation has been well-recognized in liver cirrhosis, where knowns EC stimulants such as VEGF, TNF-α, IL-6, IL-8, and IL-1ra, are significantly elevated in the plasma of patients with cirrhosis [37–39]. The severity of EC activation, and thus the remodeling of VE-Cadherin borders, is determined by integration of these pro-, and anti-inflammatory signals by the complex cell signaling pathways [40]. Here, we showed that the distribution of VE-Cadherin border morphologies correlates to the liver cirrhosis disease severity. In addition, we were able to identify that plasma from decompensated liver cirrhosis patients induces different VE-Cadherin morphologies than healthy controls and the majority of patients with compensated liver cirrhosis. Our pipeline allows improved interpretability of the results compared to classic image analysis, as the profiles represent the spread in known VE-Cadherin morphologies, contrary to interpreting Factor regression coefficients in Factor Analysis [41], a dimension reduction strategy often used in high-content screening.

It is important to stress the influence of using 25% human EDTA plasma in the patient stratification, and how this influences the comparison to know EC stimuli. Plasma, even plasma derived from healthy controls, influences VE-Cadherin border morphologies compared to the use of cell culture medium with bovine derived serum [19]. The known EC stimuli experiment were performed without a background of 25% human plasma. Furthermore, the screening of liver cirrhosis patient plasma responses has been performed on a coated glass substrate, compared to the coated plastic surface used for the screening of the EC stimulants. The choice of cell culture surface is known to influence the morphologies of VE-Cadherin and behavior of EC [42]. For optimal comparison between know titrated EC stimuli and patient plasmas, the screenings needs to be repeated on similar cell culture surfaces and EC stimuli need to be added in a background of 25% EDTA plasma.

The focus of this study was to develop a methodology and pipeline allowing fast and easily interpretable stratification of VE-Cadherin morphologies within a high-throughput screening experiment. For this reason, we only screened a selected number of known EC stimuli, and reanalyzed a previously published dataset. Our approach suffers from several limitations. Most importantly, all datasets have been acquired using the same microscope, objective lens, and camera, and our protocols have been optimized for the best image quality possible. Therefore,

the training dataset contains no variation as far as image acquisition is concerned and contains relatively little noise in the images. When analyzing lower quality or lower resolution images this might present a problem and ROIs will be classified as to contain a more diffuse VE-Cadherin border. Furthermore, this pipeline is geared towards analyzing images obtained using a high NA 20x objective lens and a 2048x2048 pixel camera. Even though image enhancement has been used for training the model to make the model less dependent on the exact size of the border, it potentially performs best using this microscope configuration. Furthermore, the image intensity normalization in script potentially suffers from the presence of debris in the images. We therefore prescreened the images to exclude images with large debris structures. However, this is not a standard procedure for most microscopy users.

Our analysis pipeline, coupled with the high-content screening of ex vivo EC morphological responses to patient plasma provides a tool to assess disease progression as well as to identify factors in the plasma that impact on the integrity of the endothelium. Ideally, our pipeline would allow to compare patient specific responses to a database with known EC stimulants, and thus identifying driving key circulating factors in the plasma and enabling tailored treatment for that specific patient. In addition, our pipeline allows to be used in inhibitor library screenings and small molecule screenings, identifying key disease drivers present in patient plasma for many systemic disease such as sepsis, heart failure, or vasoplegia.

## Author Contributions

**Conceptualization:** Rudmer J. Postma, Anton Jan van Zonneveld.

**Formal analysis:** Rudmer J. Postma.

**Writing – original draft:** Rudmer J. Postma, Susan E. Fischer.

**Writing – review & editing:** Roel Bijkerk, Anton Jan van Zonneveld.

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
