## [Decision Letter · Decision Letter 0]

3 Sep 2024

PONE-D-24-30402Unveiling Endothelial Cell Border Heterogeneity: VE-Cadherin Adherens Junction Stratification by Deep Convolutional Neural Networks.PLOS ONE

Dear Dr. Postma,

Thank you for submitting your manuscript to PLOS ONE. After careful consideration, we feel that it has merit but does not fully meet PLOS ONE’s publication criteria as it currently stands. Therefore, we invite you to submit a revised version of the manuscript that addresses the points raised during the review process.

 In particular, concerns were raised regarding the function of the code available to readers of the manuscript. It is crucial that this concern is addressed fully. Both reviewers also raised concerns regarding the interpretation and presentation of the data as well as associated statistical analyses.

We look forward to receiving your revised manuscript.

Kind regards,

Jordan Robin Yaron, Ph.D.

Academic Editor

PLOS ONE

Reviewers' comments:

Reviewer's Responses to Questions

**Comments to the Author**

1. Is the manuscript technically sound, and do the data support the conclusions?

Reviewer #1: Yes

Reviewer #2: Partly

2. Has the statistical analysis been performed appropriately and rigorously? 

Reviewer #1: Yes

Reviewer #2: No

3. Have the authors made all data underlying the findings in their manuscript fully available?

Reviewer #1: Yes

Reviewer #2: No

4. Is the manuscript presented in an intelligible fashion and written in standard English?

Reviewer #1: Yes

Reviewer #2: Yes

5. Review Comments to the Author

Reviewer #1: Postma et al. developed a neural network-based tool to differentiate VE-cadherin morphology in endothelial cells which can be attributed to stimulation by different circulatory factors. Additionally, the application of this tool in identifying disease progression in liver cirrhosis was demonstrated. The introduction and methods sections of the manuscript are described in detail. However, additional information in results and discussion sections will be helpful to the reader. The work can be helpful in diagnosis of diseases involving endothelial barrier disruption.

1. In Figure1, the authors should also include higher magnification images in inset. Also, scale bars should be included in the microscopy images.

2. Why were the specific stimulants used? What is their pathophysiological relevance in the disease progression?

3. In Figure 4, additional discussion on the effects of different treatments on the cells and their corresponding clusters is required. For example, authors mention that “Stimulation by 50 µM Histamine resulted in zig-zag focal borders to become a dominant phenotype marked by a shift towards cluster 12, 17, 18 and 19, and a loss in cluster 1.” What does loss of cluster 1 mean for cell border morphology? Also, the implications of gain or loss of individual clusters need to be discussed in terms of how they relate to the treatments.

4. In figure 6c, cosine similarity was observed between profiles of healthy subjects and patients in compensated phase in majority of cases. However, there is a considerable difference in VE-cadherin morphology in healthy and compensated groups in figure 6b. Are these images not representative? If they are, please include why the differences are not visible in pipeline developed by authors. If not, please include representative images.

5. The limitations of the tool in its current state need to be discussed.

Reviewer #2: Postma and colleagues developed a computational strategy for classification of VE-cadherin junctional morphology in images of cultured endothelial cells. Using a large training dataset of images of cells cultured under various experimental stimuli, they demonstrate that the pipeline generates clusters in UMAP embedding that represent distinct junctional morphologies. They further demonstrate that this model can be used to classify images from independent experiments (VEGF or TNFa stimulation; treatment with liver disease patient- or healthy control-derived plasma). This is a potentially very useful tool in the field of endothelial cell biology, but there are a few weaknesses that should be addressed prior to publication:

Major:

The authors’ pipeline would be of benefit to many researchers but improvements to the code are necessary to allow others to use it: In attempting to run the code provided in the authors’ Github repository (https://github.com/rjpostma/VE-Strat), I encountered several problems. I attempted this in Python v3.10.9 and using the package versions listed in the manuscript:

- embeddings_and_analysis.py, line 92, function RobustScaler generates a not defined error (should there be a “from sklearn.preprocessing import RobustScaler” command above?)

- embeddings_and_analysis.py line 93, function UMAP generates a not defined error (should there be a “from umap import UMAP” command above?)

- embeddings_and_analysis.py line 96, function HDBSCAN generates a not defined error. Here I needed to install the package hdbscan (this was not listed in the methods section of the manuscript along with the other packages) and add an import command.

- Related to the above, it would be helpful to list the required packages in the README

- embeddings_and_analysis.py, lines 70 and 76 require directories “liver_set/plate1/” and “liver_set/plate2”, which are not present in the repository (these data should be provided).

- To proceed, I changed the above directories to “TRAINING/STIMULI/Unexposed” and “TRAINING/STIMULI/IL1 high dose.” Then, embeddings_and_analysis.py, line 103 produces error “AttributeError: 'ImageXpress_filetree' object has no attribute 'stack'. And looking at the class ImageXpress_filetree in cell_border_identification_function_classes.py, there is no attribute “stack” (although there is an attribute “stacks”).

- Related to the above, in the README authors write “The pipeline we created is mostly geared towards the ImageXpress microscope filestructure, but can be adapted to other image sets as well.” It would be very helpful if authors could provide some code and guidance on using images from other microscopes.

- Code used to generate plots shown in the manuscript (e.g., UMAP, heatmaps, PCA) should be provided.

In summary, it is very important that authors provide working code that can reproduce the results in the manuscript and adequate documentation that would allow others to apply the pipeline to their own images.

In Fig. 3, there are 21 clusters present in the UMAP plot, but authors show example images for only 10 clusters and discuss only 5 in the text (clusters 1, 12, 19, 6, and 7). Could authors provide information on all clusters or provide some rationale for only considering some of the clusters? It would also be helpful if authors could comment on whether this clustering is robust to the number of junctions within an image, the angle of the junction, and other latent variables.

In Fig. 4, authors show the effects of various treatments on VE-cadherin junctional morphology as reflected by cluster occupancy. In the text authors summarize these observations for several treatments, e.g., “10 ng/ml TNFα resulted in almost complete loss of cluster 1, 6, and 7, and a shift towards cluster 12”; “Histamine resulted in zig-zag focal borders to become a dominant phenotype marked by a shift towards cluster 12, 17, 18 and 19, and a loss in cluster 1.” It would be helpful to provide a quantitative visualization (e.g., bar plots showing the % of images in each cluster for each treatment). It is also unclear whether this experiment included biological replicates (e.g., independent wells of cultured cells that underwent the same treatment). Such replication and appropriate statistical analysis is necessary to provide support for the above-mentioned claims of changed junctional morphology distribution.

In Fig. 6, authors apply their model to images of VE-cadherin junctions from cells treated with patient- or healthy control-derived plasma. They use a PCA plot to visualize cluster profiles; it would be helpful to also show the UMAP/cluster plots (analogous to those displayed in Figs. 3, 4, and 5) for at least one example from each treatment group. Based on this analysis the authors make conclusions, e.g., “Healthy control profiles also resembled profiles from compensated patients to a certain degree” and “All plasma from decompensated patients seemed to induce a VE-Cadherin morphology different from healthy controls.” The authors need to incorporate a statistical analysis to support these claims.

Minor:

In the Methods, authors write “Following image quality control, images were corrected for uneven illumination and vignetting.” Please provide details of how this correction was performed.

In Fig. 6, authors show PC1 and PC3. Could authors please explain why PC2 was omitted and provide the fraction of variance explained by the first three PCs individually?

Authors use cosine similarity to compare cluster occupancy across conditions; could authors please explain their rationale for choosing this distance metric?

In the abstract, authors write “We developed an image analysis pipeline, capable of intuitively and robustly stratify all VE-Cadherin morphologies within a sample.” This is not grammatically correct; please replace “stratify” with “stratifying.” Similarly, in the discussion, authors write “The focus of this study was to develop a methodology and pipeline allow fast and easily interpretable stratification of VE-Cadherin morphologies within a high-throughput screening experiment.” Please replace “allow” with “allowing.”

6. PLOS authors have the option to publish the peer review history of their article (what does this mean?). If published, this will include your full peer review and any attached files.

Reviewer #1: No

Reviewer #2: No

---

## [Author Response · Author response to Decision Letter 0]

16 Oct 2024

Dear editor, dear reviewers,

We thank you kindly for reviewing the manuscript, and for your questions, valuable feedback, and improvements regarding the manuscript. We have updated the manuscript, the figures, and the code on GitHub. Below, we answer the questions of the reviewers in detail one by one.

We highly appreciate your time and are looking forward to your response.

Sincerely,

Rudmer Postma

1. In Figure 1, the authors should also include higher magnification images. Also, scale bars should be included in the microscopy images.

The figure has been updated to include scalebars and a magnification of the borders.

2. Why were these specific stimulants used? What is their pathophysiological relevance in the disease progression?

We used these specific stimulants because they are often used as a model for artificial inflammation and are well described in literature. The manuscript has been updated to include a short explanation of their (patho)physiological relevance.

“TNFα and IL1β are pro-inflammatory cytokines involved in several disease and conditions including sepsis and tumor invasion, and both cytokines induce degradation of the EC barrier and expression of adhesion molecules. The growth factor VEGF is crucial for angiogenesis both physiological in wound healing as pathophysiological in many conditions including cancer and diabetic complications. EC stimulation by histamine or thrombin causes acute degradation of the EC barrier and is involved in allergic reactions and coagulation events, respectively.”

3. In Figure 4, additional discussion on the effects of different treatments on the cells and their corresponding clusters is required. For example, authors mention that “Stimulation by 50 µM Histamine resulted in zig-zag focal borders to become a dominant phenotype marked by a shift towards cluster 12, 17, 18 and 19, and a loss in cluster 1.” What does loss of cluster 1 mean for cell border morphology? Also, the implications of gain or loss of individual clusters need to be discussed in terms of how they relate to the treatments.

The effects of the treatments on the clusters are now described detail in the results section. 

“Major differences in morphology distribution between the unstimulated control and 10 ng/ml TNFα could be observed, where stimulation by 10 ng/ml TNFα resulted in almost complete loss of cluster 0, 1, 5, 7, 10, and 11, and a shift towards cluster 9 and 4, representing highly disturbed morphology. Contrary, when stimulating cells by a combination of TNFα + TGFβ there was an enrichment observed for the 11 and 12 compared to the TNFα only condition. Exposure to 0.1 ng/ml TNFα resulted mostly in enrichment of cluster 3 and 4 from cluster 0. Stimulation by IL1β resulted in a similar shift, though less severe than stimulation by TNFα. Stimulation by 50 µM Histamine resulted in zig-zag focal borders to become a dominant phenotype marked by a shift towards cluster 2, 5, 6-12, and a loss in cluster 0. Interestingly, stimulation of ECs by 100 ng/ml VEGF for two hours resulted in different distribution of morphologies when compared to overnight exposure, where two hour exposure resulted in gain in 2, 5, 6-12, and a loss in cluster 0, overnight exposure showed mostly enrichment in cluster 3 and 4 when compared to control condition, and loss 2,6,8,9 when compared to the two hour stimulation.” 

The biological meaning and implication on morphology and barrier function of the shift between the clusters is now described in the results section. 

“Our pipeline was able to discern between the different VE-cadherin morphologies, and capture this information in clusters of VE-Cadherin morphology that can be used in further data analysis as VE-Cadherin “morphological profiles”; a 1D vector containing the heterogeneity of all VE-Cadherin morphologies present in a sample. Using these profiles we were able to stratify the different stimuli. As expected, the unstimulated control condition showed higher population of clusters 0 and 1 compared to the highly stimulated conditions, representing mature borders and mature reticular areas respectively, where the 10 ng/ml TNFα condition showed highly disturbed areas, represented by cluster 3 and 4. As stimulation by high dose TNFα is associated with decreased endothelial cell border integrity, this specific morphology potentially corresponds to leaky borders. In addition, enrichment of clusters 5, 8,9, 11, and 12 as observed for stimulation by histamine likely is also associated with increased barrier permeability, as stimulation by histamine is known to induce a leaky EC phenotype. Interestingly, we also observed a time dependent effect of VEGF. As previously reported[1, 2], we observed more zig-zag like remodeling of the VE-Cadherin borders after two hours of VEGF exposure, compared to more slightly disturbed morphology after overnight exposure to VEGF. This information was also well represented in the change in clustering, and thus in the VE-Cadherin morphological profiles.”

4. In figure 6c, cosine similarity was observed between profiles of healthy subjects and patients in compensated phase in majority of cases. However, there is a considerable difference in VE-cadherin morphology in healthy and compensated groups in figure 6b. Are these images not representative? If they are, please include why the differences are not visible in pipeline developed by authors. If not, please include representative images.

Using the full multiplex of stainings within the examples makes the VE-Cadherin borders less visible. We updated the figure to only include nuclei and VE-Cadherin borders. Moreover, we included the UMAP for the example to make shifts in clusters more clear. In addition, we included scale bars in the images

5. The limitations of the tool in its current state need to be discussed.

Limitations are now included in the discussion. 

“Our approach suffers from several limitations. Most importantly, all datasets have been acquired using the same microscope, objective lens, and camera, optimized for the best image quality possible. Therefore, the training dataset contains no variation as far as image acquisition is concerned and contains relatively little noise. When analyzing lower quality or lower resolution images this might present a problem and ROIs will be classified as to contain a more diffuse VE-Cadherin border. Furthermore, this pipeline is geared towards analyzing images obtained using a high NA 20x objective lens and a 2048x2048 pixel camera. Even though image enhancement has been used for training the model to make the model less dependent on the exact size of the border, it potentially performs best using this microscope configuration. Furthermore, the image intensity normalization in script potentially suffers from the presence of debris in the images. We therefore prescreened the images to exclude images with large debris structures. However, this is not a standard procedure for most microscopy users.”

To answer the questions of reviewer number two:

Major

1. The authors’ pipeline would be of benefit to many researchers but improvements to the code are necessary to allow others to use it: In attempting to run the code provided in the authors’ Github repository (https://github.com/rjpostma/VE-Strat), I encountered several problems. I attempted this in Python v3.10.9 and using the package versions listed in the manuscript:

The problems have been resolved, and the updated code has been committed to the GitHub archive. Furthermore, functions have been included for easy display of the UMAPs and extraction of computation of cluster information. In addition, a file describing the right dependencies has been included on the GitHub archive, and a mistake in the methods section regarding the correct UMAP package has been fixed.

2: In Fig. 3, there are 21 clusters present in the UMAP plot, but authors show example images for only 10 clusters and discuss only 5 in the text (clusters 1, 12, 19, 6, and 7). Could authors provide information on all clusters or provide some rationale for only considering some of the clusters? It would also be helpful if authors could comment on whether this clustering is robust to the number of junctions within an image, the angle of the junction, and other latent variables.

To also comply with the next point raised by the reviewer, we optimized the clustering to have increased cluster size, thereby reducing the number of clusters to increase the robustness, and highly aid in interpretation/visualization of the embeddings. This also allowed us to discuss all clusters without disregarding the small clusters in the text and figure. In addition we discussed in the results section how inclusion of a global pooling layer in the model architecture helps make the method robust against variables such as the locality and orientation. 

3: In Fig. 4, authors show the effects of various treatments on VE-cadherin junctional morphology as reflected by cluster occupancy. In the text authors summarize these observations for several treatments, e.g., “10 ng/ml TNFα resulted in almost complete loss of cluster 1, 6, and 7, and a shift towards cluster 12”; “Histamine resulted in zig-zag focal borders to become a dominant phenotype marked by a shift towards cluster 12, 17, 18 and 19, and a loss in cluster 1.” It would be helpful to provide a quantitative visualization (e.g., bar plots showing the % of images in each cluster for each treatment). It is also unclear whether this experiment included biological replicates (e.g., independent wells of cultured cells that underwent the same treatment). Such replication and appropriate statistical analysis is necessary to provide support for the above-mentioned claims of changed junctional morphology distribution.

Barplots have been added to Figure 4 and 5, see figures below. Following the previous point raised by the reviewer, we increased the minimum size of the clusters to reduce the total number of clusters in order to have proper visualization when using barplots. The results section and methods section has been updated to include the proper statistics when comparing different treatments and different biological replicate conditions.

4: In Fig. 6, authors apply their model to images of VE-cadherin junctions from cells treated with patient- or healthy control-derived plasma. They use a PCA plot to visualize cluster profiles; it would be helpful to also show the UMAP/cluster plots (analogous to those displayed in Figs. 3, 4, and 5) for at least one example from each treatment group. Based on this analysis the authors make conclusions, e.g., “Healthy control profiles also resembled profiles from compensated patients to a certain degree” and “All plasma from decompensated patients seemed to induce a VE-Cadherin morphology different from healthy controls.” The authors need to incorporate a statistical analysis to support these claims.

The figure has been updated to include UMAP plots of the three given patient examples. Moreover, the statistics have been updated in the results and methods sections.

Minor:

5. In the Methods, authors write “Following image quality control, images were corrected for uneven illumination and vignetting.” Please provide details of how this correction was performed.

The Methods have been extended to include the following passage: “In short, a focus-score metric, total sum, median, standard deviation, and the 0.01, 0.25, 0.75, 0.99 quantiles of the intensity values were obtained for each image.[3, 4] Results were plotted in histograms comprised of all images for a given channel identify outliers[4] followed by inspection of images to assess image quality before discarding. Images that contained artifacts (e.g. out-of-focus images, debris, clipping/saturation artifacts), were discarded. Sites that contained failed images in a single channel were discarded altogether.

Images were corrected for uneven illumination and vignetting by using an approach was used, based on Draper et al.[5]. Bright areas or debris were iteratively filtered out by assigning areas with intensity values 3*SD above the local median pixel value the intensity value of the direct neighborhood, after which a median weighted average of the images was computed, followed by smoothing by a Gaussian filter. The image used for correcting illumination was constructed by dividing the mean background value by the smoothed artificial background image. All images in the corresponding channel and plate were multiplied by the correction image to obtain the illumination corrected images.”

6. In Fig. 6, authors show PC1 and PC3. Could authors please explain why PC2 was omitted and provide the fraction of variance explained by the first three PCs individually?

With the updated clustering and analysis, we have adjusted figure 6 to include the explained variance and PC 1 and PC 2 in the score plot.

7. Authors use cosine similarity to compare cluster occupancy across conditions; could authors please explain their rationale for choosing this distance metric?

The methods section has been update to include the rationale: “Cosine similarity as a metric was chosen because of its well described use in comparing morphological data in the field of high-content screenings[6], and because the cluster-profiles represent a frequency vector, and therefore the approach for comparing frequency vectors by cosine similarity was adopted from the field of text analysis.[7]”

8. In the abstract, authors write “We developed an image analysis pipeline, capable of intuitively and robustly stratify all VE-Cadherin morphologies within a sample.” This is not grammatically correct; please replace “stratify” with “stratifying.” Similarly, in the discussion, authors write “The focus of this study was to develop a methodology and pipeline allow fast and easily interpretable stratification of VE-Cadherin morphologies within a high-throughput screening experiment.” Please replace “allow” with “allowing.”

Thank you for drawing our attention to this. We have corrected the abstract and the discussion

---

## [Decision Letter · Decision Letter 1]

14 Nov 2024

PONE-D-24-30402R1Unveiling Endothelial Cell Border Heterogeneity: VE-Cadherin Adherens Junction Stratification by Deep Convolutional Neural Networks.PLOS ONE

Dear Dr. Postma,

Thank you for submitting your manuscript to PLOS ONE. After careful consideration, we feel that it has merit but does not fully meet PLOS ONE’s publication criteria as it currently stands. Therefore, we invite you to submit a revised version of the manuscript that addresses the points raised during the review process.Please address the final comments from Reviewer 2 regarding the presentation of data in Figure 4 and the descriptions in the methods section regarding the statistical treatment of the data.

We look forward to receiving your revised manuscript.

Kind regards,

Jordan Robin Yaron, Ph.D.

Academic Editor

PLOS ONE

Journal Requirements:

Reviewers' comments:

Reviewer's Responses to Questions

**Comments to the Author**

1. If the authors have adequately addressed your comments raised in a previous round of review and you feel that this manuscript is now acceptable for publication, you may indicate that here to bypass the “Comments to the Author” section, enter your conflict of interest statement in the “Confidential to Editor” section, and submit your "Accept" recommendation.

Reviewer #1: All comments have been addressed

Reviewer #2: (No Response)

2. Is the manuscript technically sound, and do the data support the conclusions?

Reviewer #1: Yes

Reviewer #2: Yes

3. Has the statistical analysis been performed appropriately and rigorously? 

Reviewer #1: Yes

Reviewer #2: No

4. Have the authors made all data underlying the findings in their manuscript fully available?

Reviewer #1: Yes

Reviewer #2: Yes

5. Is the manuscript presented in an intelligible fashion and written in standard English?

Reviewer #1: Yes

Reviewer #2: Yes

6. Review Comments to the Author

Reviewer #1: This is an interesting work and is potentially helpful in diagnosis of variety of diseases. The authors have written the manuscript in intelligible manner and it is easy to follow. Background, hypothesis, methods, and results are mentioned in adequate detail in the revision. Finally, authors have addressed all of my comments and the manuscript is recommended to be accepted in its revised form.

Reviewer #2: In this revised version of the manuscript, the authors have done a commendable job addressing most concerns; very importantly, I was able to run the modified code provided and generate UMAP and PCA plots. The authors have also added quantification of cluster occupancy and statistical analysis to support their claims about how VE-cadherin junctional morphology changes as a result of (a) treatment with TNFa, TGFb, VEGF, Il1B, histamine, thrombin, etc., and (b) treatment with patient- or healthy control-derived plasma.

Related to comparison (a), the figure referred to in the text when making these claims (Fig. 4) does not display quantitative data from individual replicates or a summary measure of the variance between replicates. (In contrast to comparison (b), in which each plasma sample is represented by a point in the PCA plot). Displaying replicates and/or and indicator of variance between replicates (e.g., error bars) is necessary. The authors also state in the methods “The differences between population of the clusters for different conditions were compared using Chi-square test, using a multiple technical replicates combined for the analysis.” Did authors truly combine (lump together) data from all replicates? These replicates define the variability of a given treatment and should be considered in the statistical analysis.

7. PLOS authors have the option to publish the peer review history of their article (what does this mean?). If published, this will include your full peer review and any attached files.

Reviewer #1: No

Reviewer #2: No

---

## [Author Response · Author response to Decision Letter 1]

20 Nov 2024

Dear Editor, Dear Reviewers,

We thank you kindly for the instructive feedback.

We have updated the manuscript to address the final concern of reviewer 2. Below we explain point by point how we addressed the concerns in the manuscript.

Related to comparison (a), the figure referred to in the text when making these claims (Fig. 4) does not display quantitative data from individual replicates or a summary measure of the variance between replicates. (In contrast to comparison (b), in which each plasma sample is represented by a point in the PCA plot). Displaying replicates and/or and indicator of variance between replicates (e.g., error bars) is necessary. The authors also state in the methods “The differences between population of the clusters for different conditions were compared using Chi-square test, using a multiple technical replicates combined for the analysis.” Did authors truly combine (lump together) data from all replicates? These replicates define the variability of a given treatment and should be considered in the statistical analysis.

We have altered our analysis to be able to use information from the technical replicates. We have therefore switched from using Chi-square test on the combined replicates (where we assumed combining technical replicates results in a better estimation of the true cluster belongings) to using the multivariate generalization of the T-test; the Hotelling’s t-squared test, combined with LDA analysis/visualization. Figure 4 is updated to include bars representing the standard deviating between replicates for each condition and cluster, and LDA analysis is shown to visualize how the replicates cluster together in the dimension reduced space. In addition, figure 5 has been updated to keep statistical approach the same between the two sections.

We highly appreciate your time and are looking forward to your response.

Sincerely,

Rudmer Postma

---

## [Decision Letter · Decision Letter 2]

22 Dec 2024

Unveiling Endothelial Cell Border Heterogeneity: VE-Cadherin Adherens Junction Stratification by Deep Convolutional Neural Networks.

PONE-D-24-30402R2

Dear Dr. Postma,

We’re pleased to inform you that your manuscript has been judged scientifically suitable for publication and will be formally accepted for publication once it meets all outstanding technical requirements.

Kind regards,

Jordan Robin Yaron, Ph.D.

Academic Editor

PLOS ONE

Additional Editor Comments (optional):

Reviewers' comments:

Reviewer's Responses to Questions

**Comments to the Author**

1. If the authors have adequately addressed your comments raised in a previous round of review and you feel that this manuscript is now acceptable for publication, you may indicate that here to bypass the “Comments to the Author” section, enter your conflict of interest statement in the “Confidential to Editor” section, and submit your "Accept" recommendation.

Reviewer #1: All comments have been addressed

Reviewer #2: All comments have been addressed

2. Is the manuscript technically sound, and do the data support the conclusions?

Reviewer #1: Yes

Reviewer #2: Yes

3. Has the statistical analysis been performed appropriately and rigorously? 

Reviewer #1: Yes

Reviewer #2: Yes

4. Have the authors made all data underlying the findings in their manuscript fully available?

Reviewer #1: Yes

Reviewer #2: Yes

5. Is the manuscript presented in an intelligible fashion and written in standard English?

Reviewer #1: Yes

Reviewer #2: Yes

6. Review Comments to the Author

Reviewer #1: All the comments have been addressed. The manuscript presents interesting research in an intelligible manner. The presented data support the conclusions that were drawn. This manuscript will benefit the research community. I recommend this manuscript to be accepted.

Reviewer #2: I thank the authors for addressing my final comment regarding statistical treatment of the data. This manuscript is suitable for publication in PLOS ONE.

7. PLOS authors have the option to publish the peer review history of their article (what does this mean?). If published, this will include your full peer review and any attached files.

Reviewer #1: No

Reviewer #2: No

---

## [Editor Report · Acceptance letter]

26 Dec 2024

PONE-D-24-30402R2 

PLOS ONE

Dear Dr. Postma, 

I'm pleased to inform you that your manuscript has been deemed suitable for publication in PLOS ONE. Congratulations! Your manuscript is now being handed over to our production team.

Kind regards, 

on behalf of

Dr. Jordan Robin Yaron 

Academic Editor

PLOS ONE